# communications
# engineering

# Humanoid robots to mechanically stress human cells grown in soft bioreactors

Pierre-Alexis Mouthuy [1✉], Sarah Snelling[1], Rafael Hostettler[2], Alona Kharchenko[2], Sarah Salmon[1], Alan Wainman [3], Jolet Mimpen [1], Claudia Paul[1] & Andrew Carr[1]

For more than 20 years, robotic bioreactor systems have facilitated the growth of tissue-engineered constructs using mechanical stimulation. However, we are still unable to produce functional grafts that can translate into clinical use. Humanoid robots offer the prospect of providing physiologically-relevant mechanical stimulation to grafts and implants which may expedite their clinical deployment. To investigate the feasibility of a humanoid bioreactor, we have designed a flexible bioreactor chamber that can be attached to a modified musculoskeletal (MSK) humanoid robot shoulder joint. We demonstrate that fibroblast cells can be grown in this chamber while undergoing physiological adduction-abduction on the robotic arm. A preliminary evaluation of the transcriptome of the cells after 14 days indicated a clear influence of the loading regime on the gene expression profile. These early results will facilitate the exploration of MSK humanoid robots as a biomechanically more realistic platform for tissue engineering and biomaterial testing applications.

[1] Botnar Institute of Musculoskeletal Sciences, Nuffield Department of Orthopaedics, Rheumatology and Musculoskeletal Sciences, University of Oxford, Oxford OX3 7LD, United Kingdom. [2] Devanthro GmbH, 85748 Garching, Germany. [3] Sir William Dunn School of Pathology, University of Oxford, Oxford OX1 3RE, United Kingdom. ✉email: pierre-alexis.mouthuy@ndorms.ox.ac.uk

Robotic devices are increasingly used in tissue engineering and tissue culture strategies to provide mechanical cues and modulate the growth of cells and tissue. They are typically integrated in systems called bioreactors, which control the conditions necessary for maintaining and stimulating living cells and tissues outside the body. Mechanical stresses (tension, compression, torsion and shear stresses) naturally occur in vivo and are crucial to the development and maintenance of musculoskeletal tissues. For instance, tendons (connective tissues between muscles and bones) undergo profound modifications when deprived of tensile stresses, including a reduction of their anatomical size, a rapid deterioration of their mechanical properties and a change in extracellular matrix (ECM) composition and organisation[1,2].

The application of cyclic stretching in current bioreactor systems used for tendon tissue culture has been shown to promote cell proliferation, induce tenogenic differentiation and improve matrix deposition relative to static cultures[3–6]. Torsion forces have been used to further improve bioreactor functionality and tendon construct properties[5,7]. Fundamental mechanotransduction studies have also clearly established that mechanical cues influence ECM composition and organisation in vitro. Cells sense their mechanical environment through multiple mechanisms including the interaction of integrins and syndecans with the local ECM, leading to phenotypic responses that adapt cells to their environment, via, for example, changes in gene expression[8,9]. Overall, the existing evidence suggests that delivering advanced physiologically relevant mechanical cues will improve the quality of tendon grafts; however, current bioreactor systems do not have the capacity to deliver such cues.

There are two major limitations of current tendon bioreactors that use robotic components for mechanical stimulation. First, they poorly mimic the mechanical stresses experienced by tendons in vivo. They often provide tissue constructs with uniaxial tensile stresses with a linear actuator, while multiaxial and multiple stress types are found under physiological conditions[10]. Second, the one for all approach, i.e. one bioreactor system to grow all tendons types, lacks physiological and clinical relevance. The stress distribution profile observed in tendons varies, depending on their location, size, function and on patient-specific factors (anatomy, level of physical activity, etc.). This should be considered, since applying inappropriate physical stresses may impact the structure and function of an engineered tissue graft.

The considerations above suggest that bioreactor systems that closely replicate the structures and mechanics of the human body are needed. As proposed in a perspective article in 2017, real-size musculoskeletal (MSK) humanoid robots could be a solution to achieve this[11]. MSK humanoid robots are a class of humanoids that aim to replicate the human MSK system by mimicking the inner structures of the human body such as muscles, tendons and bones[12–16]. Their actuators mimic the physiological behaviour of muscles by pulling the skeletal structure using a series of strings. Compared to the better-known humanoids such as Asimo[17] and Atlas[18], anatomically-inspired MSK humanoid robots ensure kinematics and dynamics similar to the human body, mimicking its morphology, speed and range of motions, as well as muscle force profiles. Doing so, they aim to offer safer interaction with humans. They have seen major developments in recent years, and it is now possible to explore their use for new applications in medicine, e.g. tissue engineering for clinical translation.

In the present study, we investigated the feasibility of a humanoid bioreactor in the context of tissue engineering of tendons at the shoulder joint. Torn rotator cuff tendons are the most common cause of shoulder pain in adults and affect over 25% of the population above the age of 60. Surgical repairs are commonly performed but around 40% of these repairs fail due to poor tissue healing despite advances in surgical techniques. This high failure rate adversely affects patients and supports the need for new strategies, such as engineering tendon autografts, to improve patient outcomes. We carried out this investigation using a modified MSK humanoid robotic shoulder and by designing a flexible bioreactor chamber. The chambers, containing the cell-material constructs, were positioned at the site of the supraspinatus tendon on the robotic arm and were subjected to repeated adduction/abduction motions to demonstrate the approach.

## Results and discussion

**Adapting the MSK humanoid shoulder**. The robot employed in this study was a modified version of the shoulder joint proposed by Devanthro's open-source modular MSK robotic toolkit and originally designed as a part of the biologically-inspired tendon-driven humanoid robot Roboy 2.0[19]. The original MSK shoulder, seen in Fig. 1a, b and Supplementary Movie 1, was a ball-and-socket joint actuated by up to nine artificial muscles positioned on a trunk-like stand. Each artificial muscle was a series elastic actuator using a brushless DC motor[20]. The motor coiled or uncoiled an inelastic cord travelling through three pulleys, as shown in Fig. 1c. One of the pulleys was attached to a spring-loaded guiding rod to endow the artificial muscle with elasticity and compliance. All those components were enclosed in 3D-printed polyamide casings. Since actuators could only pull, the system required antagonist and protagonist artificial muscles.

It is worth noting that this original humanoid shoulder was neither designed for medical applications nor for faithfully replicating the human shoulder. One of its main limitations was the lack of scapula, which has a crucial role in shoulder motion. As a result, this model only provided 3 degrees of freedom (DOF). In contrast, human shoulders have 6 DOF, often approximated with 4 DOF[21]. However, the 3 DOF offered by Devanthro's model were more than sufficient for this feasibility study, which focused on small abduction-adduction motions (1 DOF).

To improve the mimicry of the original shoulder model, the arm and joint structures were replaced by an assembly shown in Fig. 1d, e and Supplementary Fig. 1. The new bone structures were created by merging the CAD drawings of the original humanoid arm with the scans of human bone models (more specifically, this included the proximal humerus, glenoid socket and coracoid process of the scapula). A surgical shoulder implant was used to ensure smooth movements of the humeral head in the glenoid cavity (Fig. 1d) and an artificial ligament capsule made of elastic tape was used to hold the joint at rest (Fig. 1e). The humeral head contained a socket at the supraspinatus tendon insertion site for the bioreactor chamber to be positioned into. With these changes, the final modified MSK shoulder matched the anatomy of a human shoulder more closely than the original model. This can be appreciated qualitatively by comparing Fig. 1b, e.

The adaptation of the shoulder not only improved the clinical relevance of the model but it also suggests the potential to replicate a patient's anatomy by using clinical data, such as CT scans. Although this offers some exciting prospects, future work in this direction will require the establishment of quantitative approaches to better assess the degree of mimicry between human and humanoid structures.

**Design of a soft, flexible bioreactor chamber**. The design and components of the bioreactor chamber used in this study are shown in Fig. 2a–e. Further details can be found in Supplementary Fig. 2. The main components of the chambers included a porous aligned scaffold, a tubular membrane and rigid inserts with attached tubing. All bioreactor components used for

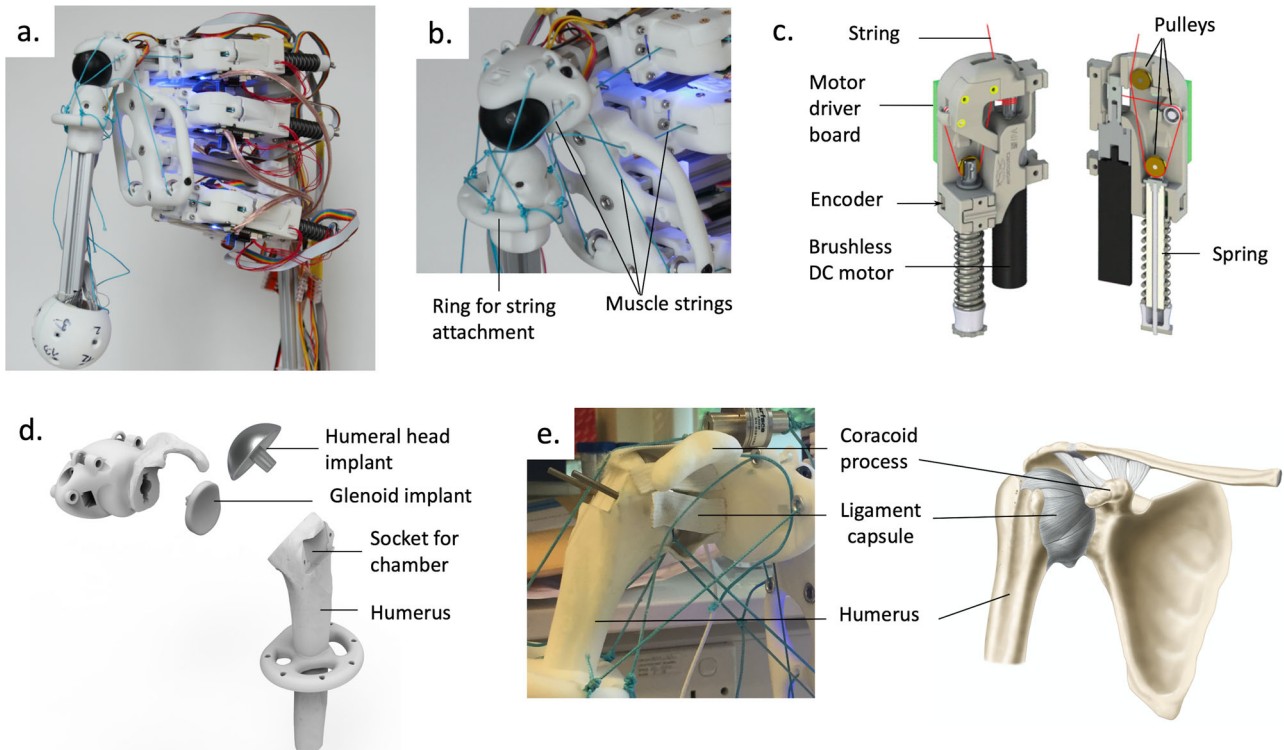

**Fig. 1 Structural adaptation made to an existing MSK humanoid shoulder. a** Original design of the humanoid arm (*Devanthro*), involving a simplified ball into socket joint and up to 9 muscle actuators, **b** close up on the original shoulder joint showing the ring used for attaching the muscle strings and the socket joint, **c** muscle actuator design (figure adapted from[46]), **d** key parts of the new shoulder design showing a 3D printed humerus (partial) and join socket (with coracoid process), fitted with the two components of a stemless shoulder implant, **e** assembled shoulder joint replacing the original design (left) showing similarities the structure of a human shoulder (right).

manufacturing the chamber were shown to be biocompatible (Supplementary Fig. 3).

The scaffold or artificial matrix was made of parallel polycaprolactone filaments produced by electrospinning. A total of 200 filaments (regrouped in 5 bundles of 40 filaments each) were stretched through the tubular membrane and fixed in the inserts at both ends with medical-grade resin. As shown in the scanning electron microscopy (SEM) images of Fig. 2c, each filament was made of aligned submicron fibres of an average diameter of 1μm. This highly anisotropic structure was designed to mimic the alignment of collagen fibres in tendons. The use of parallel filaments also led to a highly porous structure with large spaces that allow cell distribution throughout the scaffold during seeding (Supplementary Fig. 4a).

The membrane was a thin sheet of transparent polyurethane rolled into a tubular shape and sealed along the long edge. It surrounded the scaffold and acted as the chamber's wall to contain the medium and maintain sterility. A 3D printed ring and plate (Supplementary Fig. 2b, c) maintained the membrane's tubular form and secured it against the inserts with screws.

Two identical 3D printed inserts were placed on each side of the chamber. Each insert contained 5 blind channels receiving the scaffold material and a go-through channel for perfusing the medium (Supplementary Fig. 2d, e). Each of the blind channels connected to a smaller channel running perpendicularly to allow resin injection. Screw holes tapped in to the inserts allowed attachment of the plate holding the membrane. At their back, an eyelet enabled the attachment of the muscle string. Tubing was fitted in the go-though channel of each insert. These channels acted as the inlet and outlet for perfusion of the medium during culture. They were also used to seed the cells onto the scaffold material.

The resulting assembled chamber offers two major advantages compared to existing bioreactor chambers. First, it is soft and flexible: this differs radically form the rigid casing that encloses the cell-material constructs in current bioreactor systems and it enables the application of multiaxial motions by complex actuation system like humanoid robots. Torsion, compressive and tensile stresses can easily be transmitted through the non-load-bearing membrane. Second, the chamber is independent from the actuation system. In traditional bioreactors the mechanical components (actuated rods and grips) go across the chamber's walls, preventing easy separation. Here, the chamber can be detached from and attached to the robotic system at any time during the culture with the use of tubing stoppers without compromising the sterility of the tissue construct. This feature facilitates handling of the tissue construct during culture such as to carry out non-invasive monitoring of cell viability. It also makes the chamber compatible with existing commercially available actuation and mechanical testing devices.

The positioning of the chamber onto the robotic arm structure is shown in Fig. 2f: one end of the chamber was fitted into the humeral head (in the small socket created for this purpose), while the other end was attached to the top muscle string via a force sensor. This matched the supraspinatus tendon's location and recreated a bone-tendon-muscle complex (Fig. 2f, right). It is worth noting that this positioning can easily be altered (location of socket and muscle choice) if the interest is on a different tendon, highlighting the potential to perform tissue-specific experiments.

The physical properties of the chamber can be seen in Fig. 2g–j. The resistance of the membrane to pressure, tear and slippage can be appreciated in Fig. 2g and h. Characterisation of the tensile properties indicates that the membrane had a negligible

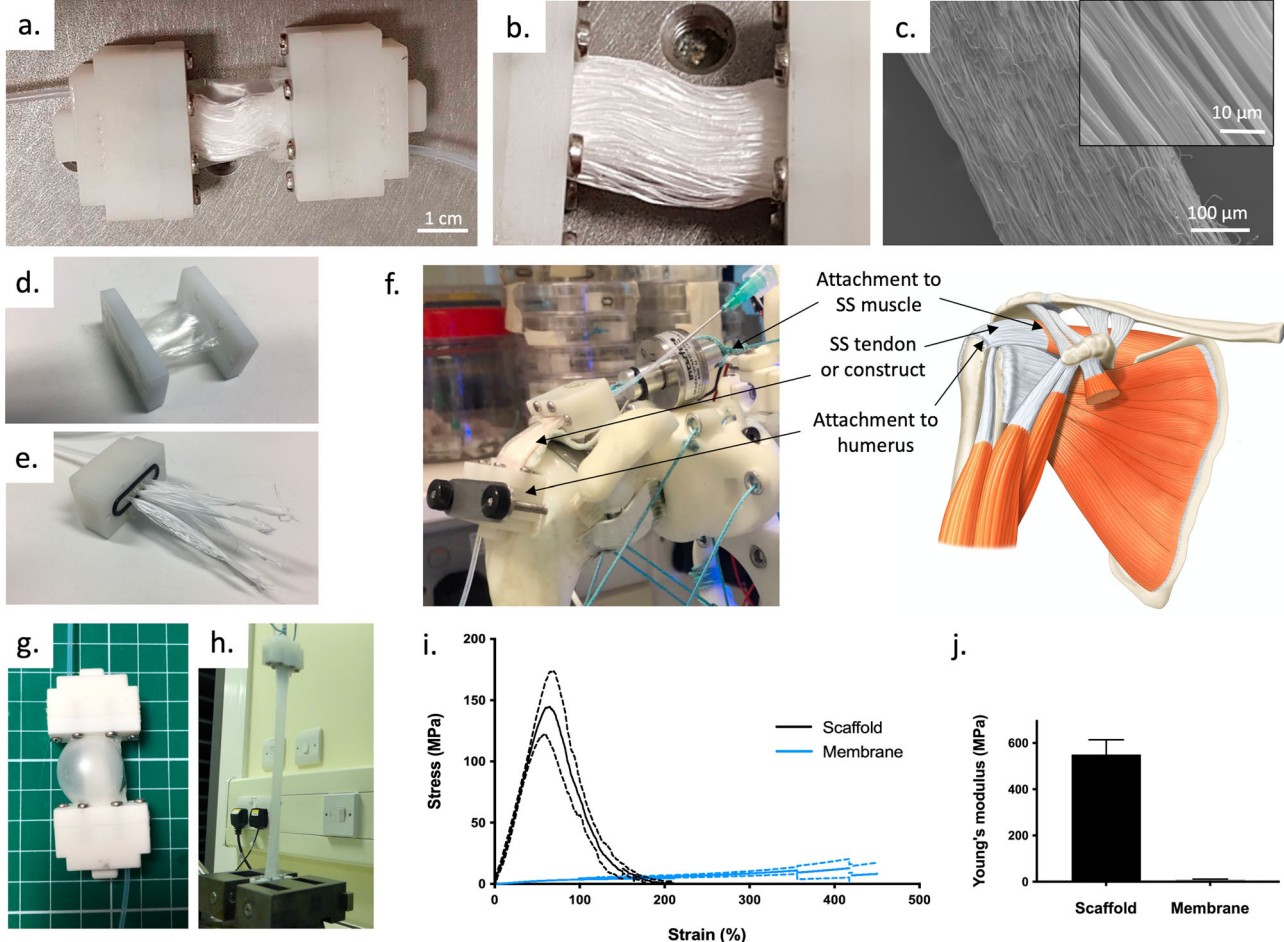

**Fig. 2 Design and properties of the flexible bioreactor chamber used in this study. a** Assembled bioreactor chamber, **b** focus on the scaffold material (membrane removed for clarity), **c** SEM images of a single filament at two different magnifications (right hand side corner: higher magnification) showing the highly aligned microfibres, **d** membrane and its holding parts, ready to be slid around the scaffold and fixed on the main insert, **e** filament bundles inserted in the blind channels of the main insert, **f** chamber positioned at the supraspinatus (SS) location on the robotic arm (left) showing the correspondence with the supraspinatus tendon location on a human shoulder (right), **g** chamber under air pressure showing expansion of the membrane (**h**) tensile test to failure of a chamber showing that the membrane keeps it integrity at high strains, **i** Average stress-strain curves of the scaffold and membrane (measured separately) with standard deviation (dashed lines) suggesting the small contribution of the membrane to stress at low strains, **j** graph comparing the Young modulus of the scaffold and the membrane (at low strains, $n = 6$, error bars represent standard deviation).

contribution to load bearing at low strains, as shown in Fig. 2i while they maintained their integrity up to $440.8 \pm 47.8\%$ of strain (also see Supplementary Movie 2, showing a complete chamber undergoing a tensile test to failure). The maximum stress observed for the scaffolds was $150.0 \pm 26.5$ MPa for a corresponding strain of $34.8 \pm 3.5\%$. The bell-shaped curve observed for the scaffold results from the successive failure of the filaments occurring after the maximum. Figure 2j. highlights the large difference between the Young's modulus of the scaffold, which was $550.6 \pm 63.4$ MPa, and the tubular membrane, which was $10.3 \pm 1.0$ MPa. Overall, the mechanical properties of the electrospun scaffold reasonably matched values typically observed for human tendon tissues[22,23].

**Mechanical stimulation of cell-material constructs on the robotic arm.** Human fibroblast cells were seeded on the scaffolds through the inlet and the chambers were then connected to the perfusion system positioned in a tissue culture incubator. This system included a peristaltic pump, an oxygenator and a reservoir bottle containing a culture medium, all connected by tubing in a closed-loop configuration (Fig. 3a–c). During perfusion, the chamber was filled with about 2 mL of culture medium, which is noticeably lower than typical volumes surrounding tissue constructs in existing bioreactor chambers (typically between 10 to 100 mL). After a 24 h rest period following seeding, the dynamic samples were transferred daily for a period of 30 min onto the humanoid arm to undergo adduction-abduction motions (Fig. 3d, e). This study was limited to the use of such motions (1 DOF, rotation in the coronal plane) because the supraspinatus tendon is mainly recruited under adduction-abduction. Also, the adaptations brought to the robotic shoulder to improve anatomical relevance introduced, as a trade-off, instabilities at higher DOF due to the smaller contact area in the joint: this made it challenging to enable reliable loading regimes. Exploring a wider range of motion (*e.g.* 3 DOF) will be the focus of future work, as it will need a robotic shoulder that includes both range of motion and anatomical relevance as design criteria.

As shown in Fig. 3f, g, two loading regimes were used: a low force regime (LFR) and a high force regime (HFR). Both regimes displayed a similar motion profile with a total motion angle of about 60° (Fig. 3f, Supplementary Movies 3 and 4). The fact that

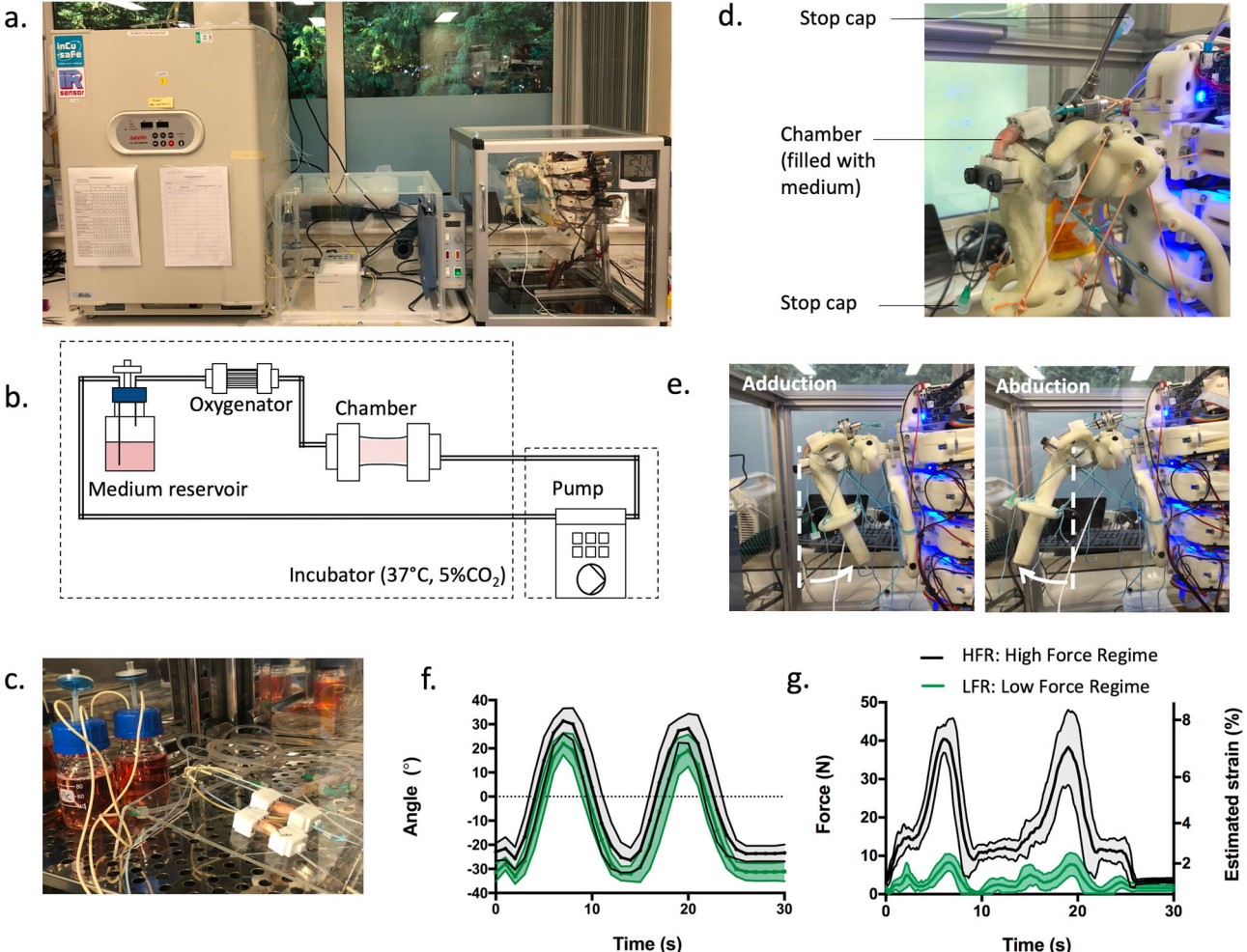

**Fig. 3 Humanoid bioreactor system and loading regimes applied during tissue culture. a** Overview of the bioreactor setup showing the incubator (left), peristaltic pump (middle), and humanoid arm (right), **b** sketch of the bioreactor perfusion circuit showing the tubing connecting the chamber to an oxygenator, a medium reservoir and the pump, **c** positioning of the chambers during rest in the incubator, **d** positioning of the chambers during mechanical stimulation on the robotic arm, **e** arm under motion showing adduction (left) and abduction (right), **f** angles measured during loading for the high force regime (HFR, black) and the low force regime (LFR, green) with 0° corresponding to the position of the arm parallel to the trunk side, **g** measured forces and estimated strains for LFR (green) and HFR (black). The upper and lower boundaries in **g** and **f** represent the standard deviation ($n = 20$).

this angle was not as wide as the recorded ones (90°) may be due to gravity and friction within the artificial muscles (*e.g.* by the string and spring), and this observation could also be made with the original shoulder joint shown in Fig. 1a. Motions were being played at the frequency of 4 arm lifts per minute, *i.e.* about 0.066 Hz, and a 5 second rest period was imposed every two-arm lifts.

As seen in Fig. 3g, LFR and HFR had a significantly different force profile. LFR showed an average $F_{max}$ of 11 N and HFR showed an average $F_{max}$ of 45 N. The range of force used here is consistent with those estimated in the supraspinatus tendon for small adduction-abduction motions[24–27]. By imposing physiological forces, the humanoid-bioreactor setup applies stresses to cell-material constructs in a similar way to what the human body does to the supraspinatus tendon or would do to an implanted material at that anatomical location. This approach therefore brings clear clinical relevance to tissue engineering and contrasts from existing bioreactor systems that are typically used to impose strains to tissue constructs. It is worth noting that during the 1DOF abduction-adduction, the supraspinatus tendon is also subjected to compression forces which contribute to maintaining

the head of the humerus in the glenoid cavity[28]. Similarly, compression of the tissue constructs can be expected with the robotic shoulder. These forces will be investigated in future work.

Strains in the scaffold material were estimated using the tensile data described earlier (Fig. 2i). At $F_{max}$, strains were calculated to be about 2.6% for the LFR and 8% for the HFR. These strain values correspond roughly to the minimum and maximum strains typically reported in dynamic tendon tissue engineering studies[29].

**Cell culture in the humanoid bioreactor system.** Figure 4a–c shows the gross appearance of the samples after 14 days of culture, following the seeding of human fibroblasts. No differences between static controls (no mechanical stimulation) and dynamic samples could be observed macroscopically. Confocal microscopy indicated the presence of cells in the static and LFR samples, while few cells could be seen in the HFR samples (Fig. 4d–i). This was consistent with the cell proliferation data shown in Fig. 4j and was further confirmed by histological staining of sample cryo-sections (Supplementary Fig. 4b–d). The fastest cell proliferation

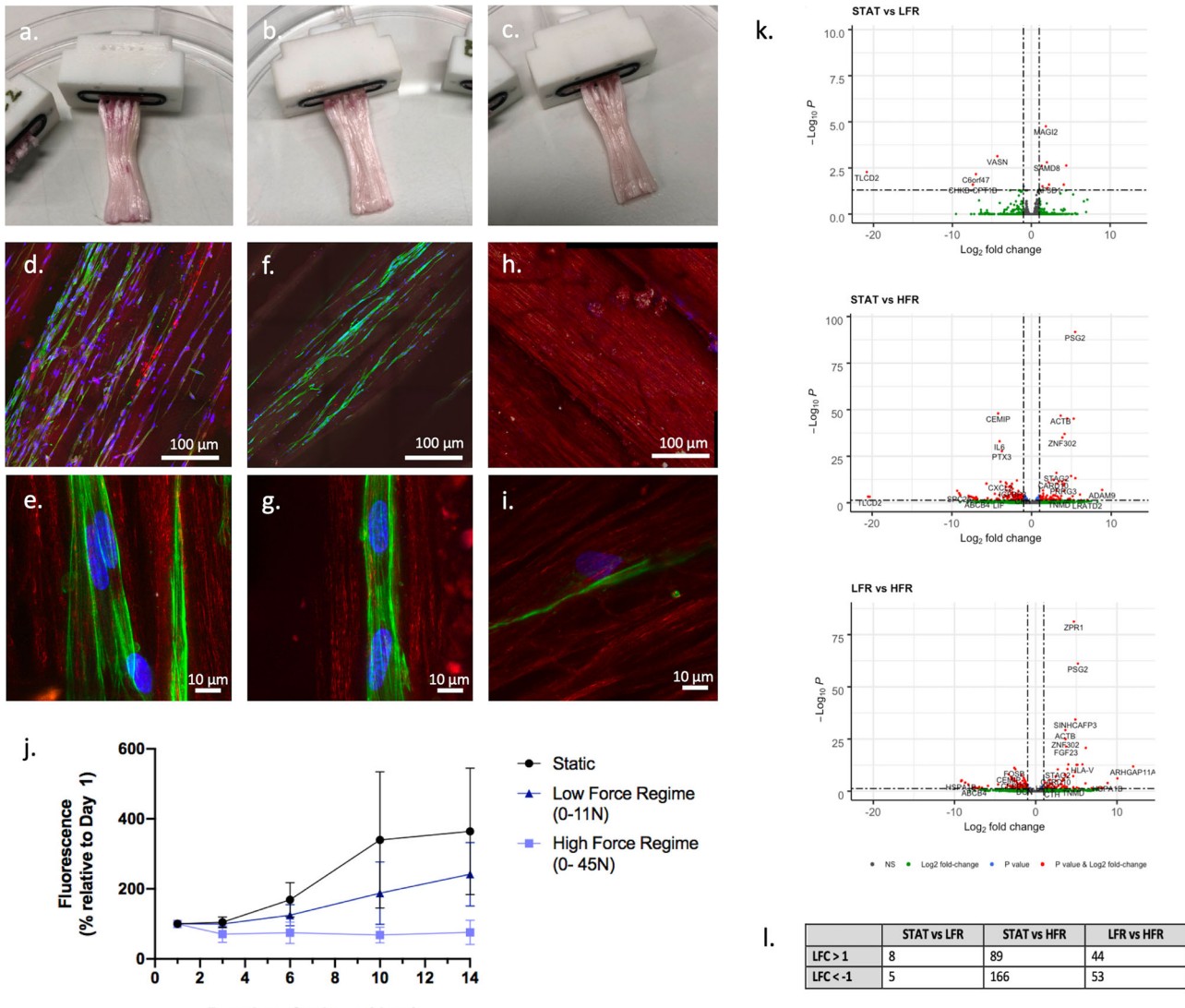

**Fig. 4 Biological characterisation of the cell-material constructs. a–c** Gross appearance of the samples following a 14 days culture period of human fibroblasts: static (**a**), LFR (**b**) and HFR (**c**); **d–i** overview and detail fluorescence images showing the cells (green: actin, blue: nuclei) in the static (**d, e**), LFR (**f, g**) and HFR (**h, i**) samples (the red signal is autofluorescence from the scaffold material); **j** Viability data obtained by Presto blue for samples culture for 14 days under the different loading regimes (static, LFR, HFR: $n = 4$ biologically independent samples, error bars represent standard deviation); **k** Differentially expressed genes for the comparisons Static vs LFR, Static vs HFR, and LFR vs HFR (static: $n = 3$; LFR, HFR: $n = 2$), **l** Overview of differentially expressed genes (padj < 0.05) for each comparison.

was seen for the static controls. A slightly slower proliferation was seen under LFR and a reduction in the cell population was observed under HFR. Cells displayed a highly elongated morphology regardless of the condition applied.

A preliminary evaluation of the transcriptome of the fibroblast cells at 14 days was also carried out using bulk RNA-Seq. After normalisation, log-fold2 changes (LFC) were calculated with the adjusted p-value (padj)<0.05. Differentially expressed genes (padj < 0.05, LFC ± 1) are shown using volcano plots in Fig. 4k. While differentially expressed genes were found between static and LFR samples (13 genes), more changes were found between static and HFR samples (255 genes) and LFR and HFR samples (97 genes), indicating a clear influence of the loading regime on the gene expression profile (Fig. 4l). The 10 most significantly changed genes are depicted in heatmaps in Supplementary Fig. 5.

Overall, these early experiments indicate the feasibility of using flexible bioreactor chambers in combination with humanoid robots to apply mechanical stimulation on growing cells. The

work demonstrates that cells can be maintained within the bioreactor system for extended periods (14 days) and that they respond to loading. This provides a strong platform for future investigation of a wider range of motions (e.g. 2 or 3 DOF) and of loads (larger forces), utilising cells derived from human tendons, to address how it can improve on traditional bioreactors. We observed low cell viability under HFR (strain ~8%) and few statistically significant changes in gene expression profile under LFR (strain ~2%). This contrasts with traditional uniaxial bioreactor systems that show proliferation at strain up to 10% and more variation in gene expression profile at low strain, in particular for genes associated with ECM production[6,30,31]. Potential reasons for these discrepancies are explained in the paragraphs below.

The reduction in cell numbers observed under HFR (0-45 N) suggests that cells underwent apoptosis or detached from the scaffold in response to the high mechanical stresses, and this may have several explanations. First, while electrospun polycaprolactone

has been shown to encourage cell growth under load, the stiffness of our scaffold (550 MPa) was much higher than the scaffolds tested in other studies (typically < 100 MPa)[32–34]. Such stiffness is mainly a consequence of the pre-stretching needed to prevent plastic deformation during use and to create an ECM-like fibre arrangement that can induce an elongated cell morphology[35,36]. The high stiffness may have resulted in local shear stresses that were unfavourable for cell survival at HFR. Softer scaffolds, which better mimic the mechanical properties of tendon at the different hierarchical levels may reduce cell stress and improve viability. Another reason is that loadings were applied very early following seeding (at day 1). Progressive loading regimes (e.g. mimicking rehabilitation exercises), giving more time for cells to generate a more suitable extracellular matrix environment, reduce shear stress and to establish cell-cell connections for better communication are likely to improve the ability of cells to withstand HFR conditions. A more prolonged period of rest at the start of the experiment could be beneficial[4]. Finally, as mentioned before, the cell-material constructs are likely to have experienced compressive stresses under adduction-abduction, unlike when subjected to uniaxial loading in traditional bioreactors. These additional stresses may also influence cell viability[37].

While the low viability indicates that cells become stressed under HFR, cells proliferated under LFR and showed significant changes in gene expression compared to static samples. The genes affected in this study are not those typically reported in previous bioreactor work, such as type I collagen, tenascin C, tenomodulin and scleraxis, where the aim has been to generate tendon-like constructs[4,31,38]. A dermal fibroblast cell line was used for this preliminary work, not tendon-derived fibroblasts, and as such changes in tendon-like genes were not anticipated and comparisons regarding changes in tendon-related genes cannot be made. The relatively low number of gene-expression changes compared to studies utilising traditional uniaxial bioreactors may represent the physiological response to such low-force loading within a multiaxial system. However, this response may also be contributed to by the high stiffness of the scaffold, compressive forces, and uneven tensions between the filaments following fixation in the resin (with only a few stiff filaments possibly taking the load).

Future work will explore the previously discussed aspects in depth to ensure that the humanoid bioreactor better matches the performance of traditional bioreactors in terms of viability and changes in gene expression. Exploring various scaffold materials, more similar to those previously used, will be needed. Involving a wider range of loading regimes including controls such as uniaxial stretch (equivalent to traditional bioreactor) and 3 DOF motion (daily tasks, abnormal tasks) will also be particularly important to improve our understanding of this new culture platform. Furthermore, monitoring the environment in the chamber during culture using sensors to measure temperature, major nutrients and metabolites (oxygen, glucose, lactate, etc.), and local stresses and strains will also support our understanding of the system and the effect of conditions applied. The development of solid and fluid computational models alongside will be key to support this experimental work. Finally, involving tendon cells donated from patients, using more biological replicates, and carrying out further biological and biomechanical characterisation of the tissue constructs will be crucial to assess the function of future tendon tissue constructs.

**Potential implications of this new bioreactor strategy**. Here, we demonstrate the feasibility of using MSK humanoid robots to support tendon tissue engineering by growing cells in a flexible bioreactor chamber, which can be mechanically stimulated on a humanoid robotic arm. Although, as mentioned before, further work is necessary to assess the full potential of this strategy, it has the ability to overcome the existing limitations of current bioreactor systems. By using robotic structures to replicate the complex mechanics of the human body, it enables physiological and clinical relevance from a mechanical point of view. Supraspinatus tendon tissue engineering is used as a pilot application; however, a similar approach could be extended to engineer other tendons (i.e. for different locations in the body) or even other tissues (bone, ligaments, muscles, etc.). MSK humanoids also offer the prospect of personalising mechanical stimulation during in vitro tissue generation: while the morphology of the robot can be made to match that of the patient, the kinetics and kinematics could be personalised as well. Advances in artificial intelligence research will contribute to this, in addition to providing real-time monitoring and enabling adaptive behaviour during culture. With such a wide potential, the use of MSK humanoids could lead to transformative steps in tissue engineering and regenerative medicine. Developments in tissue engineering currently overlook the need for advanced bioreactor systems. This may be a prerequisite for the fabrication of constructs that are fully functional, preventing the translation of engineered grafts into clinics. Access to functional engineered tissue grafts in clinics will positively impact the quality of life of patients (faster and better repair), on society (faster return to work and to social activities) and on the economy (lower healthcare costs).

Besides tissue engineering applications, humanoid bioreactor systems may become advanced in vitro culture models to test cells, drugs and biomaterials. Studying cells and tissues under more realistic mechanical stress will lead to better understanding of biological processes. This could in turn improve knowledge in medical pathology, clinical diagnosis, and treatment strategies, including refining rehabilitation programs[39]. These applications could reduce the use of animals in preclinical trials by preventing the translation in vivo of strategies involving biomaterials that are not mechanically appropriate. While high throughput screening can be achieved on organ-on-chips systems[40], models working at the real tissue and organ scales with physiologically-relevant mechanical loadings are an important step in the development of new healthcare strategies.

Despite the various advantages that a humanoid bioreactor may offer, it is important to note that it does not intend to replace existing dynamic platforms. Instead, it aims to fill a gap in the translational pathway to clinics that has not yet been addressed. Among limitations, a humanoid bioreactor approach implies low throughput (each chamber requiring its own perfusion loop), single-use chambers, highly mechanically competent scaffolds, a less robust and user-friendly robotic platform, as well as higher complexity and cost to run experiments.

The development and demonstration of usefulness of humanoid bioreactors will be tightly dependent on progress in robotics, as future work will benefit from more stable robotic systems (to safely explore higher DOF) that are even more closely mimicking the biomechanics of the human body. For instance, shoulder models involving scapula motion will be particularly important to obtain structures and performances similar to human shoulders and to explore various loading regimes informed by physiologically relevant situations (including rehabilitation exercises)[41]. Humanoid bioreactors may in turn stimulate the development of humanoids by generating new opportunities in medicine and biomedical research. Furthermore, future work in this direction may support the development of biohybrid robots and cell-based actuators[42–45], such as by providing a flexible environment (soft chamber) able to maintain muscle cells for long periods of time.

## Conclusions

We have proposed an approach to tissue engineering by integrating MSK humanoid robots into a bioreactor platform. To demonstrate feasibility, we adapted an existing humanoid shoulder and developed a soft, flexible chamber that can be hosted by the robot. This proof-of-concept bioreactor platform can maintain cells in culture and deliver load that induces changes in gene expression and proliferation. Future work should investigate the effect of various loading regimes, scaffold materials, cell types and operating parameters. Possible long-term benefits from a humanoid bioreactor-based strategy include the production of functional tissue grafts for patients, the creation of an improved in vitro culture model for preclinical work and the opportunity to support the development of advanced robotic systems.

## Materials and Methods

**Original humanoid shoulder**. The original robotic shoulder used in this study was built by General Interfaces GmbH (Garching, Germany) based on the open-source designs of Devanthro's Roboy Project (https://devanthro.com/technology/). In brief, the model contained 6 working robotic muscles (one was used as a spare) mounted on a stand made of aluminium rods (20x20mm extrusion profile) and 3D printed parts. Each muscle included a 100 W brushless motor fitted with a 1:53 Gearbox (Maxon Motors AG, Sachseln, Switzerland), an encoder, a spring and a series of pulleys[46]. These components were contained in a 3D printed casing designed with a CAD software (Autodesk Fusion, San Francisco, United States) and made of 3D printed polyamide (EOS GmbH Electro Optical Systems, Krailling, Germany). A motor driver board was used to control the attached motor and to read the encoder. The encoder was used to discern the motor position and spring deflection, which were directly related to the force applied to the tendon string (300 lb Hercules Polyethylene Super Tackle 8 Strands Strong Braid Fishing Line, Hercules, Taiwan). The motor driver board was used to send this data via an SPI interface to a controller board and to receive PWM signals to control the motor. The control loop ran at 2 kHz. The controller board was based on a DE10-Nano-SoC development board, which combined a Cyclone V FPGA with an ARM Cortex-A9 processor. A PID controller was running on the FPGA to control motor position, spring-displacement (related to force) or motor velocity. It used Ubuntu 16.04 with ROS Kinetic as the middleware. A GUI based on rviz allowed to record the muscle trajectories and replay them. In the shoulder model, the muscles were used to actuate an arm made of an aluminium rod ending with a spherical plastic knob. The strings from the muscles were attached to a 3D printed ring fixed onto the aluminum rod. The ball of the arm was fitted into a 3D printed shoulder socket attached to the main aluminium frame.

**Modifications of the robotic arm**. The structure of the shoulder model was modified by merging the existing CAD engineered parts (General Interfaces GmbH, Garching, Germany) with the 3D scans of scapula and humerus bone models (AnatomyStuff, Health Books UK Ltd, Aberystwyth, UK). The geometries of bone models and of a shoulder implant were scanned with a 3D scanner (Faro, Rugby, UK) and reverse engineered using Geomagic Design X (3Dsystems, High Wycombe, UK) and ANSYS Spaceclaim (ANSYS Inc, USA) to generate a 3D model. These models were then merged with the existing CAD drawings of the original robotic shoulder. From the scapula, only the glenoid and the coracoid process features were maintained and features were added to the glenoid to enable the attachment of a keeled glenoid implant made of polyethylene (Zimmer biomet, Indiana, USA). On the humerus side, only the proximal part was kept and the head was cut to leave space for a stemless humeral head implant made of a cobalt chrome alloy (Zimmer biomet, Indiana, USA). A socket for attaching the chamber was also created near the insertion area of the supraspinatus tendon. The final models were printed by SLS with polyamide resin (3Dsystems, High Wycombe, UK). The printed parts were then assembled together with the implant and the joint was maintained together with knitted elastic bands (Vancool, Guangdong, China), acting as artificial ligaments.

**Fabrication of the scaffold**. Electrospinning solutions were prepared by dissolving polycaprolactone (Ashland Specialities Ireland, Tallaght, Ireland) into 1,1,1,3,3,3-hexafluoroisopropanol (Apollo Scientific Ltd., Cheshire, UK) at a concentration of 16.5% (weight to volume ratio). The solution was agitated at room temperature on a roller for at least 24 h to allow for the complete dissolution of the polymer. Electrospinning was performed according to a method described previously[35]. Briefly, a custom electrospinning apparatus with a single nozzle and a stainless steel wire collector (100μm in diameter) were used to fabricate continuous electrospun filaments. The solution feed rate was 1 mL h$^{-1}$, wire feed rate was 0.5 mm s$^{-1}$ and the filaments spun under a voltage of 7.2 kV in average. The filaments were detached from the collecting wire and then stretched up to 7 times their initial length, to prevent later deformation and align the submicron fibres in the direction of the thread. The average diameter of the electrospun fibres in the drawn filaments was 1.03 ± 0.3 μm. Following stretching, bundles of filaments were fabricated by assembling 40 filaments of 5 cm in a parallel arrangement. The ends of the bundle were tied with a single filament both ends. The bundles were kept at room temperature in a desiccator until their attachment in the bioreactor chamber.

**Fabrication of the chamber 3D printed chamber parts**. Parts were drawn with Solidworks (edition 2016, Dassault Systèmes SolidWorks Corporation, Waltham, USA) and were 3D printed by selective laser sintering in polyamide 12 (3D Life-Prints, Liverpool, UK). The parts included a main insert (where the filaments were anchored), an intermediate ring (for holding the membrane) and an end plate (to secure the membrane on the main insert). Rubber o-rings (Polymax, Bordon, UK) were inserted in the groves created in the main body and in the inside of the endplate. This ensured that the chamber was leak-tight.

**Fabrication of the flexible tubular membrane**. The tubular membranes were produced from a transparent polyether-based thermoplastic polyurethane film (TFL-2EA, thickness: 50μm, kindly donated by Permali Gloucester Limited, Goucester, UK). The film was folded in two and the long edge was welded with an electric heat sealer (Cole-Palmer, Illinois, USA) to achieve an internal width of 16 mm. The tubes obtained were cut to a length of 6 cm. Each end of the tubular membrane was positioned onto an intermediate ring by feeding it through the hole and bringing its inside surface out, around the printed material. Each of the rings was then positioned into an endplate, by passing them through the plate hole. The final length of the tubular membrane stretched between the two endplates was set to approximately 30 mm.

**Assembly of the bioreactor chamber**. Assembly started by inserting the ends of the filament bundles (approximately 1 cm) in each of the 5 channels of a first insert. Epoxy resin (Epotek 301, Epoxy Technology Inc., Billerica, USA) was then injected in the channels at relatively high viscosity to avoid capillary effect in the filament bundles. The resin was left for 24 h to set. Following this, the free ends of the 5 filament bundles were passed through the tubular membrane (and its supportive parts) and positioned in the channels of the second insert and epoxy resin was injected and left to set for 24 h again. Polytetrafluoroethylene (PTFE) tubing (1/16 OD, ID 0.8 mm, Sigma Aldrich, Dorset, UK) was added to both ends of the chamber in their dedicated go-through channels and the epoxy resin was also injected around the connection to ensure a tight fit. A small loop of braided cord (10lb-300lb PE braided line, Hercules, China) was added to one end of the chamber (back channel) to enable attachment to a load cell, itself connected to the top muscle string. A summary list of the materials used in the soft chamber fabrication is given in Supplementary Table 1.

**Cell preparation and cell culture**. Human dermal fibroblast cell line (HFF-1, ATTC, Manassas, Virginia) were grown in 150 cm$^2$ flasks (Greiner, Germany) in Dulbecco's Modified Eagle Medium F12 containing 15% foetal bovine serum (Biosera UK) and 1% penicillin-streptomycin solution. Flasks were incubated at standard conditions (37 °C, 5% CO$_2$) and growth medium was replaced every 2–3 days. Cultures were maintained under these conditions until the flasks reached confluence.

Prior to seeding, all chambers were sterilized using 70% ethanol for 2 h, washed with PBS three times and dried under sterile conditions at room temperature for 3 days. Cells were trypsinised and scraped from the confluent flasks and re-suspended in culture medium. About $0.5 \times 10^6$ cells were resuspended in 250 μl of TrueGel3D (TRUE1-1KT, Sigma Aldrich, UK) and were injected in the chamber using 1 mL syringes with blunted needles (G18, Temuro Ltd, Surrey, UK). After 20 min, 1 mL of fresh medium was added to the chambers and after 1 h, the chambers were connected to the perfusion system.

**Connection of the chambers to the perfusion system**. Following seeding, the chambers were taken into a tissue culture incubator maintained at 37°C and 5% CO$_2$. The chambers were connected to the peristaltic pump (model IPC-N ISM937C, Ismatec, Wertheim, Germany) and oxygenator (Type S, Alpha Plan GmbH, Radeberg, Germany) and a medium reservoir (100 mL borosilicate glass bottle with modified caps) using PharmaMed BPT and silicone peroxide tubes (Cole Parmer, St Neots, UK), adapter (PP, male luer to 1/16" barbed, VWR International Ltd, Lutterworth, UK) and blunted needles (G18, Temuro Ltd, Surrey, UK). The pump was placed outside the incubator and the medium was perfused at a rate of 1.5 mL min$^{-1}$.

**Stimulation of the cell-material constructs on the robotic arm**. Every day during the culture period, the chambers were exercised on the humanoid arm. Prior to loading the chambers on the humanoid shoulder, a set of two adduction and abduction movements (approximately 30° each side from a reference position parallel to the trunk of the robot) were applied manually to the arm mounted with an unseeded chamber at a frequency of about 0.07 Hz. The adduction-abduction

exercises were repeated for 30 min. The tension in the spring of the supraspinatus muscle (actuator attached to the chamber) was adjusted to achieve a set maximum force (either 11 N or 45 N, depending on the regime). Once the loading regimes were set, the chambers were disconnected from the perfusion system (stop caps were placed on both ends of the chamber to maintain sterility) and transferred onto the humanoid robot located outside the incubator. About 1 mL of medium was left in the chamber to ensure that cells had enough nutrients during stimulation. Motions of the robotic arm with the bioreactor chamber were recorded with a digital camera (12MP, 30fps; Sony, Tokyo, Japan) fixed onto the front incubator wall, centred with the glenohumeral joint and parallel to the coronal plane. Fiji (ImageJ, National Institute of Health, Bethesda, MD, USA) was then used to manually measure the angles during the movements of the arm at a frequency of 1 Hz. The vertical trunk stand of the robot was used as the reference.

**Cell viability assay**. To assess cell viability in the chamber, a PrestoBlue (Invitrogen, Paisley, UK) assay was used. Culture media was removed from the chambers and replaced with 1000 μL of 10% PrestoBlue solution (v/v in DMEM/F-12). After 1 h of incubation at 37 °C, 1000 μL of PrestoBlue medium samples from each chamber was collected in sterile Eppendorf tubes and 100 μL transferred to a white 96-well plates (Corning, UK) for analysis. Fluorescence was measured using the FluoStar Optima microplate reader (BMG Labtech, Ortenberg, Germany, $\lambda_{ex} = 544$ nm, $\lambda_{em} = 590$ nm). The chamber was rinsed with 1000 μL of fresh standard medium before being reconnected to the bioreactor. The assay was performed on days 1, 3, 6, 10 and 14.

**Force measurement and strain estimation during stimulation**. A load cell (model WMC-450N; miniature type load cell, capacity 450 N, Interface Force Measurements Ltd, Crowthorne, UK) was used in combination with a portable, high-speed data logger (model 9330-MSDI-IP43, Interface Force Measurements Ltd) to record the forces applied to the tissue construct. The load cell was positioned between the muscle and the chamber and was attached by high-performance cord (10lb-300lb PE braided line, Hercules, China) on both sides. Estimated strains were calculated from the tensile force-strain graphs obtained for the scaffolds in wet conditions.

**Tensile testing**. Components of the chamber (tubular membrane and scaffold) were tested with a Zwick tensile machine (Zwick Roell Group, Ulm, Germany) at a rate of 25 mm min$^{-1}$ until failure. The samples were hydrated with phosphate buffer saline solution 2 h prior to the test. Force and elongation were recorded and the ultimate stress (MPa), Young's modulus (MPa) and breaking strain (%) were then calculated ($n = 6$). Stress was calculated as the force divided by the initial cross-sectional area. To estimate the initial cross-sectional area of the scaffold, each of the 200 filaments was considered to be 57μm in thickness and 246μm in width (average measurements), with a porosity of 60%[47]. For the membrane, a thickness of 50μm and a total perimeter of 4 cm (including the sealed edge) were used.

**Scanning electron microscopy (SEM)**. Before imaging, scaffolds were mounted on an aluminium stub using a carbon adhesive disk and gold-coated using a sputter coater (SC7620 mini, Quantum Design, Switzerland). High-resolution images were taken using an environmental SEM (Evo LS15 Variable Pressure SEM, Carl Zeiss, Germany).

**Bulk RNA Sequencing**. Constructs were harvested in 500 μl of trizol (Sigma–Aldrich, Dorset, UK). Samples were centrifuged at 13,000 rpm for 8 min at 4°C and the supernatant collected. Ribonucleic acid (RNA) was extracted using Zymo Research Clean and Concentrator kits (Cambridge Biosciences, UK) according to the manufacturer's protocol. RNA concentrations in the eluted samples were determined using a Nanodrop Machine (Implen GmbH, München, Germany). Library preparation was done using a NEBNext Ultra II Directional RNA Library Prep Kit for Illumina with poly-A selection (Illumina, San Diego, CA, USA) following the manufacturer's instructions. 100 ng of RNA was used for library preparation, apart from 3 samples (2 HFRs, and 1 LFR) where there was not 100 ng available so the maximum available amount (23.4, 26.0, and 27.3 ng, respectively) was used; two additional replication cycles were used at the end of the protocol to adjust for this. DNA content of every library was quantified using a BioAnalyzer (Agilent, Santa Clara, CA, USA). Libraries were pooled and run on an Illumina NextSeq 500 using the 75 cycles NextSeq High Output kit (Illumina).

Raw FASTQ files containing reads were generated by the Illumina software CASAVA v1.8. The raw FASTQ files were processed using workflows developed by the Cribbs' lab (https://github.com/cribbslab)[48]. The quality of the reads was assessed using FASTQC and ReadQC. Raw reads were aligned to the GRCh38 reference genome using Kallisto (v0.46.1, Pachter Lab 2019). The mapped reads were visualized using IGV (v2.3.74) to further assess the quality of mapping. The quantification of mapped reads against GCRh38 reference genome annotation was carried out using FeatureCounts (v1.5.0). Two samples (p22 HFR and p24 LFR) had to be excluded due to high percentage of over-represented sequences and low alignment scores. Downstream analyses were performed using R version 4.1.0

(R Foundation, Vienna, Austria), and RStudio version 1.4.1717 (RStudio, Boston, MA, USA). Differential expression analysis was performed using the DESeq2 package[49] using wald tests and the 'apeglm' method to apply the shrinkage of logarithmic fold change[50]. The adjusted p-value (padj) and significance of changes in gene expression were determined by applying the Bonferroni-Hochberg correction of 5% false discovery rate. PCA plots were generated using the package ggplot2[51], heatmaps were generated using the package pheatmap (CRAN 1.0.12) (PCA, heatmap, and sample-to-sample distance plots in Supplementary Fig. 6), and EnhancedVolcano (Bioconductor 3.13) was used to create volcano plots. Heatmaps were created using GraphPad Prism 9.2.0 (GraphPad Software, La Jolla, CA, USA).

**Histology**. Tissue constructs were removed from the chamber and were stored in 10% buffered formalin. They were embedded in Optimal Cutting Temperature compound (Fischer Scientific, Loughborough, UK) and sectioned at 8-10μm in a cryostat (OTF 5000, Bright Instruments Ltd, Luton, UK). After being cryosectioned, samples were stained by hematoxylin and eosin staining (Sigma-Aldrich) and slides were imaged using an upright microscope (Axio Imager M1, Zeiss, Oberkochen, Germany).

**Confocal microscopy**. The samples containing cells were fixed in 10 % formalin (Fischer Scientific, Loughborough, UK) for 30 min at room temperature before being permeabilised with 0.1% Triton X-100 (Thermo Fisher Scientific, Waltham, MA, USA) at room temperature for 5 min. Samples were then stained with acti-stain 488 phalloidin (Cytoskeleton Inc., Denver, CO, USA) and 4′, 6-diamidino-2-phenylindole (DAPI) (Thermo Fisher Scientific, Waltham, MA, USA). Samples were rinsed in phosphate-buffered saline solution (Sigma-Aldrich) and imaged at 21°C using an inverted Zeiss 880 microscope fitted with an Airyscan detector. Samples were imaged using both Plan-Apochromat 10x/0.45NA (overview) and Plan-Apochromat 63×/1.4NA (detail) lenses. 405 nm solid-state, 488-nm argon and 561-nm diode lasers were used to excite DNA, 488 Phalloidin and scaffold autofluorescence, respectively (sequential excitation of each wavelength). For overview images, z-stack with 5 μm interval were collected, in a 3×3 tile. For detail images, z-stack at 0.5 μm interval were collected. Images (showing maximum intensity projections) were then Airyscan processed (auto strength) and stitched in Zen black software.

**Statistics and reproducibility**. For all quantitative analyses, data are presented as mean ± standard deviation. For statistical comparison, t-tests or two-ways ANOVA were performed to identify significance between each group. Graphpad PRISM version 7 software (GraphPad Software Inc., La Jolla, CA, USA) was used for all statistical analysis. Statistical significance was determined at $p < 0.05$. Tensile properties of the soft bioreactor chamber were performed with $n = 6$, loading regimes applied during tissue culture were described with $n = 20$ and biological experiments were performed with $n = 4$.

**Reporting summary**. Further information on research design is available in the Nature Research Reporting Summary linked to this article.

## Data availability

Datasets and CAD files are available from the authors and are deposited on the University of Oxford's institutional repository, ORA-Data (DOI: 10.5287/ bodleian:9eX14oddB). The open-source designs of Devanthro's Roboy Project can be found at https://devanthro.com/technology/.

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

## Acknowledgements

We would like to acknowledge the assistance James Fisk and David Salisbury from the IBME workshop. We thank the Gentle Giant Studios UK team from 3Dsystems including Huseyin Caner, Jet Cooper, Elliot Viles and Henry Mountain, for their support with the 3D scanning. We also thank Permali Gloucester for kindly donating the polyurethane membranes. We acknowledge the contributions of Nicole Dvorak and Luka Savić in the production of electrospun filaments used in this study, of Iain Sander for his support in troubleshooting the robotic arm, and of Dr Adam Cribbs for his advice on the transcriptome analysis. The authors gratefully acknowledge financial support from the EPSRC heathcare technologies, grant number EP/S003509/1, the National Institute for Health Research (NIHR) Oxford Biomedical Research Centre (BRC), the Medical Science Division, University of Oxford (Pump Priming fund) and the Micron Oxford Advanced Bioimaging Unit (Wellcome Trust grant number 107457).

## Author contributions

P.M.: original idea, adaptations to robotic arm, chamber design, experimental design, sample characterization (excepted gene expression analysis), write up; S.Sn: cell experimental design, write up; A.W.: confocal microscopy, write up; S.Sa: cell culture experiments, viability data, cryosectionning; J.M. and C.P.: RNA extraction, sequencing and analysis, write up; R.H. and A.K.: robotic expertise, original robotic arm design, support with software and hardware, write up; A.C.: clinical expertise, support during developments, write up.

## Competing interests

PM is the inventor, and AC and SS are contributors, of a patent application which relates to the bioreactor chamber design (patent applicant: University of Oxford, application number PCT/GB2020/052301, status of application: PCT stage). However, there is no financial implications linked to this application that may have influenced the work described in the submitted work. All other authors declare no competing interesting.
