## [Peer Review File · Communications Engineering]

Reviewers' comments:

Reviewer #1 (Remarks to the Author):

The manuscript proposes a technically fascinating approach for mechanical stimulation in tissue-engineering using humanoid bioreactor. The authors showed that the humanoid bioreactor could perform physiological adduction-abduction while allowing cells growing. Despite the advanced technology, the feasibility demonstration was limited in one DOF stimulation, which demerits the proposed technology. The manuscript thus requires a major revision to fully demonstrate the capability of the robotics arm.

Major points:

1. While the proposed system can perform 3 DOF motion of the artificial shoulder, this manuscript demonstrated only 1 DOF stimulation on the flexible bioreactor chamber. The authors should explain how such a stimulation different from that generated by the traditional method.
2. Can the authors compare the properties of the synthesized tendon using the new method with those cultured by the traditional method?
3. It is clearly shown that the mechanical stimulation reduced the cell population under low force regime although the bioreactor was stimulated in 1 DOF. Engaging 3DOF motion of the shoulder for the stimulation may reduce the cell population further. It is necessary to synthesize the tendon under 3 DOF stimulation and compare with those produced under 1 DOF stimulation. If it is not possible, the authors should provide sufficient explanation and discussion.

Minor points:

1. The authors did not cite figure S1 in the manuscript.
2. It would be clearer if the authors can provide a list of materials for the chamber.
3. Since the proposed technology would have a significant contribution in tissue-engineering, I recommend the authors to make the hardware and software open access so that it can be widely used in the field. However, it is up to the authors to make them public.

Reviewer #2 (Remarks to the Author):

The study described in the manuscript proposes and demonstrate the use of shoulder joint of a humanoid robot as the mechanical stimulator for the human cells loaded on fibres of chamber (tendon bioreactor) attached to the shoulder joint. The main purpose is to put the cell being cultured under mechanical stress same as in vivo tendon of real human. The idea is very interesting and useful to mimic, reproduce and investigate the cell culturing in vivo. The reviewer likes the authors to consider and clarify the followings before publication.

- 1) The title is not clear or misleading. Readers cannot promptly understand what was done by using humanoid robot and how.
- 2) Line 108 "the range of motions offered by the 107 model was sufficient for this feasibility study" How to prove or justify it?

3) The paragraph from Line 110 “To improve the mimicry of the original shoulder model, ...”
How to evaluate the mimicry? It is better to quantitatively show that the modified MSK shoulder matches the anatomy of a human shoulder. Otherwise, readers cannot follow what was improved and how much.

4) Line 211, the authors assume the friction is significant in the system. Which parts of the system would cause friction? If they are nearby the chamber, the temperature of the chamber temporarily rises, and it affects the cell culture even though the chamber is in incubator. The temperature of the chamber should be monitored.

5) As for Figure 3 F, how did the authors measure the angle? The angle value technically varies depending on the camera angle to the plane of interest. It should be stated how to set the camera and to define the plane as well.

6) Better to analyse the data of Figure 4 more. As mentioned in Line 65, the stress distribution profile varies in tendon. Thus, the stresses between the fibers in the chamber should vary, and the result of Figure 4 should be different between the fibers.

7) Only adduction-abduction movement was test in the study. A few more different movements should be applied for testing.

8) “Static” is the only experiment control to the adduction-abduction movement. It is better to apply abnormal shoulder movements which are not seen in real human as control.

Reviewer #3 (Remarks to the Author):

In this study, the authors used impressive engineering to create a novel bioreactor system to mimic anatomically-accurate shoulder motions to provide more physiological loading regimens to tendon tissue engineered constructs. The bioreactor is innovative, albeit quite complex, and does address some limitations of current bioreactor systems. While I agree, that complex systems such as this have their place in translating tissue engineering therapies, there are a number of points with the current system that require further clarification in the manuscript.

While this system can better replicate the complex loading regimens found in the shoulder, the cell response to the applied loads does not suggest that appropriate anabolic strain levels were applied to the cells seeded on the scaffold. The LFR regimen only significantly affected expression of 13 genes via RNA-seq (and not the typical anabolic factors such as SCX, TNMD, COL1A1, etc?) and the HFR regimen was detrimental to cell viability. As the authors noted, the loading regimens were in ranges consistent with previous literature yet they did not find a similar anabolic response. This tempers my excitement for this system but can likely be worked out in future studies. Nonetheless, the authors should address these limitations in the manuscript.

A challenge with all tensile bioreactors, including this system, is knowing how much of the applied strain at the grips is experienced by the cells. The scaffold in this study is significantly stiffer than most PCL electrospun scaffolds (modulus of ~500MPa vs. <100MPa in previous studies), which I’m assuming is because the current scaffolds were pre-stretched (lines 391-392) prior to cell seeding.

Therefore, grip-to-grip strains measured in this system likely lead to significantly higher local strains on the cells than previous aligned PCL bioreactors. As a result, the applied loads were not beneficial to the cells.

The manuscript would also greatly benefit from a paragraph in the discussion that considers the limitations of this system compared to others in the literature and on the market. For instance, while the current system may better replicate physiologic loading patterns, it loses the high throughput of other systems and also significantly increases the level of complexity. While these different systems have their place in different stages of the translational pipeline, it would help the reader to better understand the advantages/disadvantages and use cases for this system compared to previous systems in the past.

Answers to Reviewer's comments

We would like to thank the referees for taking the time to review this manuscript and for their constructive feedback.

Reviewer #1:

The manuscript proposes a technically fascinating approach for mechanical stimulation in tissue-engineering using humanoid bioreactor. The authors showed that the humanoid bioreactor could perform physiological adduction-abduction while allowing cells growing. Despite the advanced technology, the feasibility demonstration was limited in one DOF stimulation, which demerits the proposed technology. The manuscript thus requires a major revision to fully demonstrate the capability of the robotics arm.

Major points:

1. While the proposed system can perform 3 DOF motion of the artificial shoulder, this manuscript demonstrated only 1 DOF stimulation on the flexible bioreactor chamber. The authors should explain how such a stimulation different from that generated by the traditional method.

Thank you for this comment, we agree that we are not yet exploiting the possibility of 3 DOF that the system may offer. The main reasons for this choice are as follow:

- we were focused on the supraspinatus tendon, which is mainly recruited in abduction/adduction motions (1 DOF, rotation in the coronal plane).
- the adaptations brought to the robotic shoulder to improve anatomical relevance introduced, as a trade-off, instabilities at higher DOF due to the smaller contact area in the joint. This made it challenging to enable reliable loading regimes and increased the risk of damaging samples undergoing stimulation at higher DOF. We are currently working on a system upgrade to tackle this issue.
- also, given the complexity of this novel approach, keeping things simple where possible was key to demonstrate the feasibility that humanoid robots could be used as platform for mechanically stressing cell-material constructs.

Sentences have been added to the text (lines 234-240, p7-8) in reference to this:

“This study was limited to the use of such motions (1 DOF, rotation in the coronal plane) because the SS tendon is mainly recruited under adduction-abduction. Also, the adaptations brought to the robotic shoulder to improve anatomical relevance introduced, as a trade-off, instabilities at higher DOF due to the smaller contact area in the joint: this made it challenging to enable reliable loading regimes. Exploring a wider range of motion (e.g. 3 DOF) will be the focus of future work, as it will need a robotic shoulder that includes both range of motion and anatomical relevance as design criteria.”

It is worth noting that despite applying a 1DOF movement, the tissue construct itself was experiencing a motion more complex than the 1D uniaxial one typically applied in traditional bioreactor. Abduction-adduction indeed involves a rotation around an axis (the glenohumeral joint), leading to both tensile and compressive forces. Sentences in reference to this have been added in line 260-261, p8, and lines 325-328, p10-11.

2. Can the authors compare the properties of the synthesized tendon using the new method with those cultured by the traditional method?

To answer this question and similar comments made by Reviewer 2 and 3, we have expanded the discussion in p10-11 (lines 299-356):

“Overall, these early experiments indicate the feasibility of using flexible bioreactor chambers in combination with humanoid robots to apply mechanical stimulation on growing cells. The work demonstrates that cells can be maintained within the bioreactor system for extended periods (14 days) and that they respond to loading. This provides a strong platform for future investigation of a wider range of motions (e.g. 2 or 3 DOF) and of loads (larger forces), utilising cells derived from human tendons, to address how it can improve on traditional bioreactors. We observed low cell viability under HFR (strain ~8%) and few statistically significant changes in gene expression profile under LFR (strain ~2%). This contrasts with traditional uniaxial bioreactor systems that show proliferation at strain up to 10% and more variation in gene expression profile at low strain, in particular for genes associated with ECM production [6, 31, 32]. Potential reasons for these discrepancies are explained in the paragraphs below.

The reduction in cell numbers observed under HFR (0-45N) suggests that cells underwent apoptosis or detached from the scaffold in response to the high mechanical stresses, and this may have several explanations. First, while electrospun PCL has been shown to encourage cell growth under load, the stiffness of our scaffold (550MPa) was much higher than the scaffolds tested in other studies (typically <100MPa) [33-35]. Such stiffness is mainly a consequence of the pre-stretching needed to prevent plastic deformation during use and to create an ECM-like fibre arrangement that can induce an elongated cell morphology [36, 37]. The high stiffness may have resulted in local shear stresses that were unfavourable for cell survival at HFR. Softer scaffolds which better mimic the mechanical properties of tendon at the different hierarchical levels may reduce cell stress and improve viability. Another reason is that loadings were applied very early following seeding (at day 1). Progressive loading regimes (e.g. mimicking rehabilitation exercises), giving more time for cells to generate a more suitable extracellular matrix environment, reduce shear stress and to establish cell-cell connections for better communication are likely to improve the ability of cells to withstand HFR conditions. A more prolonged period of rest at the start of the experiment could be beneficial [4]. Finally, as mentioned before, the cell-material constructs are likely to have experienced compressive stresses under adduction-abduction, unlike when subjected to uniaxial loading in traditional bioreactors. These additional stresses may also influence cell viability [38].

While the low viability indicates that cells become stressed under HFR, cells proliferated under LFR and showed significant changes in gene expression compared to static samples. The genes affected in this study are not those typically reported in previous bioreactor work, such as type I collagen, tenascin C, tenomodulin and scleraxis, where the aim has been to generate tendon-like constructs [4, 32, 39]. A dermal fibroblast cell line was used for this preliminary work, not tendon-derived fibroblasts, and as such changes in tendon-like genes were not anticipated and comparisons regarding changes in tendon-related genes cannot be made. The relatively low number of gene-expression changes compared to studies utilising traditional uniaxial bioreactors may represent the physiological response to such low-force loading within a multiaxial system. However, this response may also be contributed to by the high stiffness of the scaffold, compressive forces, and uneven tensions between the filaments following fixation in the resin (with only a few stiff filaments taking the load).

Future work will explore the previously discussed aspects in depth to ensure that the humanoid bioreactor better matches the performance of traditional bioreactors in terms of viability and changes in gene expression. Exploring various scaffold materials, more similar to those previously used, will be needed. Involving a wider range of loading regimes including controls such as uniaxial stretch (equivalent to traditional bioreactor) and 3 DOF motion (daily tasks, abnormal tasks) will also be particularly important to improve our understanding of this new culture platform. Furthermore, monitoring the environment in the chamber during culture using sensors to measure temperature, major nutrients and metabolites (oxygen, glucose, lactate, etc.), and local stresses and strains will also support our understanding of the system and the effect of conditions applied. The development of solid

and fluid computational models alongside will be key to support this experimental work. Finally, involving tendon cells donated from patients, using more biological replicates, and carrying out further biological and biomechanical characterisation of the tissue constructs will be crucial to assess the function of future tendon tissue constructs.”

3. It is clearly shown that the mechanical stimulation reduced the cell population under low force regime although the bioreactor was stimulated in 1 DOF. Engaging 3DOF motion of the shoulder for the stimulation may reduce the cell population further. It is necessary to synthesize the tendon under 3 DOF stimulation and compare with those produced under 1 DOF stimulation. If it is not possible, the authors should provide sufficient explanation and discussion.

Thank you for the comment, we agree and hope that the added explanation and discussion in response to comment #1 and #2 have provided a response to this. We believe that more work is needed before investigating a wider range of motions, such as the design of a robotic shoulder that is more stable at 2 and 3 DOF (this is ongoing work in collaboration with Devanthro, our robotic partners). The last paragraph of the discussion in p11 (lines 343-356) now highlights the future work that we intend to carry out.

Minor points:

1. The authors did not cite figure S1 in the manuscript.

This has now been corrected (line 124, p4).

2. It would be clearer if the authors can provide a list of materials for the chamber.

We have created a list of the materials used in the fabrication of the chambers and have added it as a supplementary table (Table S1). This is cited in line 528, p17 and shown in the supplementary materials file.

3. Since the proposed technology would have a significant contribution in tissue-engineering, I recommend the authors to make the hardware and software open access so that it can be widely used in the field. However, it is up to the authors to make them public.

We agree, this was indeed our intention and we have clarified this in the newly added data availability section toward the end of the manuscript (p23):

“Data availability

Datasets and CAD files are available from the authors and will be deposited on the University of Oxford’s institutional repository, ORA-Data. The open-source designs of Devanthro’s Roboy Project can be found at <https://devanthro.com/technology/>.”

Reviewer #2:

The study described in the manuscript proposes and demonstrate the use of shoulder joint of a humanoid robot as the mechanical stimulator for the human cells loaded on fibres of chamber (tendon bioreactor) attached to the shoulder joint. The main purpose is to put the cell being cultured under mechanical stress same as in vivo tendon of real human. The idea is very interesting and useful to mimic, reproduce and investigate the cell culturing in vivo. The reviewer likes the authors to consider and clarify the followings before publication.

1) The title is not clear or misleading. Readers cannot promptly understand what was done by using humanoid robot and how.

Thank you for this comment, we have now changed the title to *“Humanoid robots to mechanically stress human cells grown in soft bioreactor”*. We hope that this better reflect the manuscript.

2) Line 108 “the range of motions offered by the 107 model was sufficient for this feasibility study”. How to prove or justify it?

We appreciate that this statement may have been confusing and have reworded the sentence as follow: *“However, the 3 DOF offered by Devanthro’s model were more than sufficient for this feasibility study, which focused on small abduction-adduction motions (1 DOF).”* (lines 120-121, p4)

3) The paragraph from Line 110 “To improve the mimicry of the original shoulder model,” How to evaluate the mimicry? It is better to quantitatively show that the modified MSK shoulder matches the anatomy of a human shoulder. Otherwise, readers cannot follow what was improved and how much.

Thank you for this interesting point. While we agree that a quantitative description would have been a better way to describe mimicry, at this stage we can only provide a qualitative description. The answer to this is indeed not straightforward as it would require an important (human) reference dataset (e.g. volume/surface/length/density of bones, volume/shape/stiffness/insertion zones of connective tissues, muscle location/density/volume, etc.), the subsequent measurements of all those features on the robot and a scoring system.

We have reworded and added sentences in p4 to take into account this comment:

Lines 131-133: *“With these changes, the final modified MSK shoulder matched the anatomy of a human shoulder more closely than the original model. This can be appreciated qualitatively by comparing Fig1. B and E.”*

Lines 137-139: *“The adaptation of the shoulder not only improved the clinical relevance of the model but it also suggests the potential to replicate a patient’s anatomy by using clinical data, such as CT scans. Although this offers some exciting prospects, future work in this direction will require the establishment of quantitative approaches to better assess the degree of mimicry between human and humanoid structures.”*

We hope that this is now acceptable.

4) Line 211, the authors assume the friction is significant in the system. Which parts of the system would cause friction? If they are nearby the chamber, the temperature of the chamber temporarily rises, and it affects the cell culture even though the chamber is in incubator. The temperature of the chamber should be monitored.

Thank you for this comment. The friction that we referred to was the one caused by the moving parts within the artificial muscles (e.g. spring and string). We have clarified this in

line 245-246, p8. Moderate friction also occurred during sliding of the free end of the chamber (3DP insert) on the 'scapula bones' during abduction-adduction, but heat transfer from this material is expected to be very low due the low thermal conductivity of polyamide ($\sim 0.25 \text{ W}\cdot\text{m}^{-1}\cdot\text{K}^{-1}$). Direct friction with the membrane of the chamber was negligible as it was protected by the 3DP insert during motion.

We agree however that monitoring the chamber temperature would be desirable as it could vary during transfer of the chamber to the humanoid arm. We have added the following sentence to highlight this (lines 342-345, p11): *"Furthermore, monitoring the environment in the chamber during culture using sensors to measure temperature, major nutrients and metabolites (oxygen, glucose, lactate, etc.), and local stresses and strains will also support our understanding of the system and the effect of conditions applied."*

5) As for Figure 3 F, how did the authors measure the angle? The angle value technically varies depending on the camera angle to the plane of interest. It should be stated how to set the camera and to define the plane as well.

Thank you, we have reworded the text in lines 568-573, p17-18 to answer this comment: *"Motions of the robotic arm with the bioreactor chamber were recorded with a digital camera (12MP, 30fps; Sony, Tokyo, Japan) fixed onto the front incubator wall, centred with the glenohumeral joint and parallel to the coronal plane. Fiji (ImageJ, National Institute of Health, Bethesda, MD, USA) was then used to manually measure the angles during the movements of the arm at a frequency of 1Hz. The vertical trunk stand of the robot was used as the reference."*

6) Better to analyse the data of Figure 4 more. As mentioned in Line 65, the stress distribution profile varies in tendon. Thus, the stresses between the fibers in the chamber should vary, and the result of Figure 4 should be different between the fibers.

We agree with this comment but find it challenging to build a strong discussion around this at this stage, due to the lack of evidence. We have made a reference to this potential effect in lines 340-341, p11. We have also highlighted (lines 350-353) the need for using sensors to monitor local stresses and strains and the need for developing computational tools to support our understanding of the system.

7) Only adduction-abduction movement was test in the study. A few more different movements should be applied for testing.

Thank you for this comment, please refer our answer to comment 1, Reviewer 1.

8) "Static" is the only experiment control to the adduction-abduction movement. It is better to apply abnormal shoulder movements which are not seen in real human as control.

This is a very interesting suggestion for future work and we have added a sentence in reference to this in lines 347-348, p11. While we agree more controls would have been beneficial, the static control was necessary in the work carried out so far to assess the influence of the dynamic regimes applied (LFR and HFR).

Reviewer #3:

In this study, the authors used impressive engineering to create a novel bioreactor system to mimic anatomically-accurate shoulder motions to provide more physiological loading regimens to tendon tissue engineered constructs. The bioreactor is innovative, albeit quite complex, and does address some limitations of current bioreactor systems. While I agree, that complex systems such as this have their place in translating tissue engineering therapies, there are a number of points with the current system that require further clarification in the manuscript.

While this system can better replicate the complex loading regimens found in the shoulder, the cell response to the applied loads does not suggest that appropriate anabolic strain levels were applied to the cells seeded on the scaffold. The LFR regimen only significantly affected expression of 13 genes via RNA-seq (and not the typical anabolic factors such as SCX, TNMD, COL1A1, etc?) and the HFR regimen was detrimental to cell viability. As the authors noted, the loading regimens were in ranges consistent with previous literature yet they did not find a similar anabolic response. This tempers my excitement for this system but can likely be worked out in future studies. Nonetheless, the authors should address these limitations in the manuscript.

Thank you, we have re-worked and expanded the discussion in p10-11 with this particular comment in mind (see answer to Comment 1, Reviewer 1). It now includes potential explanations of our observations, various limitations of the study and future directions.

A challenge with all tensile bioreactors, including this system, is knowing how much of the applied strain at the grips is experienced by the cells. The scaffold in this study is significantly stiffer than most PCL electrospun scaffolds (modulus of ~500MPa vs. <100MPa in previous studies), which I'm assuming is because the current scaffolds were pre-stretched (lines 391-392) prior to cell seeding. Therefore, grip-to-grip strains measured in this system likely lead to significantly higher local strains on the cells than previous aligned PCL bioreactors. As a result, the applied loads were not beneficial to the cells.

This is a great point, we have included text in the discussion of p10-11 to highlight this as a potential reason for low cell viability. We strongly suspect that the current scaffold design is a major reason for the poor cell performances compared to traditional bioreactors (rather than the bioreactor platform itself) and ongoing work is focusing on this aspect.

The manuscript would also greatly benefit from a paragraph in the discussion that considers the limitations of this system compared to others in the literature and on the market. For instance, while the current system may better replicate physiologic loading patterns, it loses the high throughput of other systems and also significantly increases the level of complexity. While these different systems have their place in different stages of the translational pipeline, it would help the reader to better understand the advantages/disadvantages and use cases for this system compared to previous systems in the past.

We have now added a paragraph in the general discussion to highlight the limitations of this system in comparison with traditional bioreactors. This can be found in lines 404-409, p13: *"Despite the various advantages that a humanoid bioreactor may offer, it is important to note that it does not intend to replace existing dynamic platforms. Instead, it aims to fill a gap in the translational pathway to clinics that has not yet been addressed. Among limitations, a humanoid bioreactor approach implies low throughput (each chamber requiring its own perfusion loop), single-use chambers, highly mechanically competent scaffolds, a less robust and user-friendly robotic platform, as well as higher complexity and cost to run experiments."*

REVIEWERS' COMMENTS:

Reviewer #1 (Remarks to the Author):

The humanoid bioreactor proposed in this manuscript is a fascinating approach for mechanical stimulation in tissue engineering. Although there are limitations that did not enable higher DOF motions, the system already showed a huge advantage in anatomical relevance stimulation in 1 DOF (adduction-abduction) which is suitable for synthesizing the supraspinatus tendon. This new approach, particularly humanoid bioreactor, would not only promote further advances in tissue engineering but also be interesting for the wide range of audiences of Communications Engineering. The authors already addressed the issues that I raised and this revised version of the manuscript is nicely written and organized. Thus, I would like to recommend this manuscript for publication.

Reviewer #2 (Remarks to the Author):

No further comment.

Reviewer #3 (Remarks to the Author):

I would like to thank the reviewers for addressing my comments in the revised manuscript.